# From Covert Hiding To Visual Editing: Robust Generative Video Steganography

## ABSTRACT

Traditional video steganography methods are based on modifying the covert space for embedding, whereas we propose an innovative approach that embeds secret message within semantic feature for steganography during the video editing process. Although existing traditional video steganography methods excel in balancing security and capacity, they lack adequate robustness against common distortions in online social networks (OSNs). In this paper, we propose an end-to-end robust generative video steganography network (RoGVSN), which achieves visual editing by modifying semantic feature of videos to embed secret message. We exemplify the face-swapping scenario as an illustration to demonstrate the visual editing effects. Specifically, we devise an adaptive scheme to seamlessly embed secret messages into the semantic features of videos through fusion blocks. Extensive experiments demonstrate the superiority of our method in terms of robustness, extraction accuracy, visual quality, and capacity.

## CCS CONCEPTS

• **Security and privacy → Security services**.

## KEYWORDS

steganography, Video steganography, Robust video steganography

## 1 INTRODUCTION

Steganography is the science and technology of embedding secret messages into natural digital carriers, such as image [6, 41, 45], audio [11, 27], video [13, 24, 30], text [1, 9, 30], etc., which is a crux of covert communication system. Different from the message concealment of cryptography, steganography focuses on concealing the existence of secret messages. Generally, the natural digital carriers are called "cover" and the digital media with secret message are called "stego". Conventional image steganography methods [41, 43, 45] primarily modify high-frequency components to embed secret message. They commonly utilize methodologies such as pixel value manipulation or integrating secret message into the cover image before inputting it into an encoder for Steganography.

In the past few years, as the rise of short video software applications like TikTok, YouTube, Snapchat, etc., video has become a suitable carrier for steganography. Traditional video steganographic

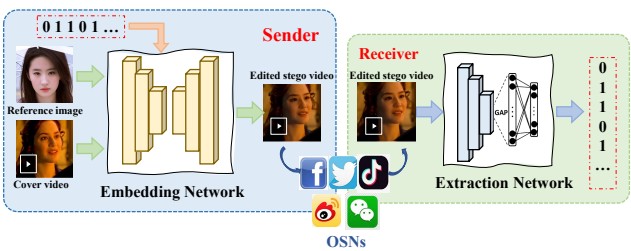

**Figure 1: Methodology of RoGVSN. We modulate semantic feature with secret message to edit videos, such as the identity feature in facial videos. Our RoGVSN can generate high-quality stego videos even in the presence of various distortions.**

methods, utilizing direct pixel value manipulation [5], coding mapping [24], or adaptive distortion function [17], exploit video data redundancy for information hiding. While exhibiting commendable security and embedding capabilities, these methods on modifying covert space can be erased by common post-processing operations easily. So they are vulnerable to mitigate diverse distortions that may occur in lossy channel transmission. These shortcomings emphasize the urgent need for further research to enhance the resilience of these methods, ensuring the reliable concealment and transmission of sensitive information in practical settings.

Visual editing on videos can be seen as the process of modifying the semantic information of objects within them. Instead of hiding secret message in covert space, we embed secret message within semantic feature of videos for visual editing. The high-level semantic feature is less susceptible to distortions, making this method inherently robust. In order to improve the robustness of video steganography, we propose an end-to-end robust generative video steganography network (RoGVSN), which consists of four modules, containing information encoding module, secret message embedding model, attacking layer, and secret message extraction module. For evaluation purpose, we employ face-swapping technology as a representative example to demonstrate the effectiveness of the proposed method. Fig. 1 illustrates the application scenario of RoGVSN. By leveraging secret messages to modulate semantic feature, such as facial identity information in videos, RoGVSN can achieve steganography during the process of visual editing. The applicability of our RoGVSN can be readily extended to various other applications. Comprehensive experiments demonstrates that our method outperforms existing state-of-the-art techniques. Notably, our approach exhibits commendable robustness against various forms of distortions and possesses strong generalization capabilities across diverse scenarios and datasets. These experiments underscore the efficacy and versatility of our method in addressing challenges in the field of visual editing and steganography.

The main contributions of our proposed method are as follows,

- We are the first to explore a novel generative video steganography method, which modifies semantic feature to embed secret message during visual editing instead of modify the covert space. This framework exhibits strong extensibility, serving as a new topic for the future development of the steganography field.
- The proposed method is robust against common distortions in social network platforms and the secret message can be extracted accurately.
- Our method achieves better security than other state-of-the-art methods, which can effectively evade the detection of steganalysis system.

## 2 RELATED WORK

### 2.1 Image Steganography

Traditional image steganography techniques predominantly involve making alterations to embed secret message on the pixel domain. The Least Significant Bits substitution method [6] operates under the assumption that human eyes cannot perceive changes in the least significant bit of pixel values. HUGO [18] is a highly secure steganography system designed to minimize distortion to high-dimensional multivariate statistics. Syndrome-Trellis Coding (STC) [14] utilizes predefined embedding costs for all pixels or discrete cosine transform (DCT) coefficients. A pioneering advancement, HiDDeN [45] introduces an end-to end trainable framework through an encoder-decoder architecture based on deep networks. SteganoGAN [43] employs dense encoders to enhance payload capacity. Volkhonskiy et al. [33] demonstrates promising performance in both the authenticity of generated images and resistance to steganalysis systems. With the development of generative adversarial networks, a wave of work on generative steganography has emerged. For instance, Wei et al [37] propose an advanced generative steganography network that can generate realistic stego images without using cover images. IDEAS [26] disentangles images into structural and texture vectors, subsequently embedding secret message into the structural vector. However, these image steganography methodologies can be obliterated by common post-processing operations, such as JPEG compression or Gaussian Blur.

### 2.2 Video Steganography

Early video steganography approaches frequently involved direct modification of the RGB or YUV color space to embed secret message. For instance, Cetin et al. [5] compute frame-specific histogram features and established a threshold for secret information embedding within RGB pixels. Subsequently, Dong Y et al. [13] discover that within the HEVC codingmodel, altering the intra-frame mode primarily affects video coding efficiency, while modifying the multilevel recursive coding unit does not significantly amplify this distortion's impact. Liu et al. [24] utilize diamond-shaped coding to enhance the expressive capability of the PU division pattern for concealed information, effectively boosting the payload. They also replace the loop filter of the I frame with a CNN to enhance the reconstruction quality of compressed images. PWRN [22] refine

this approach by employing a super-resolution convolutional neural network with a wide residual-net filter to replace the loop filter in HEVC. More recently, He et al. [17] devise an adaptive distortion function based on improved Rate Distortion Optimization and adopted Syndrome-Trellis Code [35] steganography coding to minimize embedding distortion. They also propose a super-resolution CNN with Non-Local Sparse Attention-net Filter to replace the loop filter in HEVC, thereby reconstructing the reference frame and enhancing visual quality. However, these methodologies exhibit limited robustness, rendering the concealed secret message susceptible to inadvertent loss during subsequent compression coding. Moreover, they grapple with the challenge of mitigating the impact of diverse distortions that may arise during lossy channel transmission. Inspired by semantic-based image steganography [26, 44] schemes, we propose RoGVSN, a novel approach that robustly transmits secret message through the modification of identity feature within facial videos.

### 2.3 Visual Editing

Visual editing encompasses a diverse array of manipulations, ranging from basic adjustments to complex transformations. Such manipulations can include color correction to enhance or modify the appearance of individual elements within the images [29], as well as the deletion, addition [40], or alteration of objects to either enhance the composition or convey a different narrative [38]. Additionally, techniques like image blending or compositing enable the seamless integration of multiple photographs to generate composite images with enhanced aesthetic or communicative value. In the realm of videos, visual editing extends to a broader range of manipulations, including the application of effects to specific frames to enhance visual impact or convey specific emotions. Furthermore, it involves the removal or addition of elements within the video to alter the narrative or aesthetic presentation. Notably, face-swapping techniques have gained prominence, allowing for the replacement of one person's face with another's, thereby enabling novel creative possibilities or facilitating various applications such as identity protection or entertainment [7].

## 3 METHODOLOGY

### 3.1 Overview

In this section, we will describe the framework of our proposed RoGVSN. Our method aims to embed secret message $M$ using semantic feature extracted from the reference image $I_R$ into the cover video $V_C$, generating the stego video $V_C^{'}$. Subsequently, various distortions such as video compression, noise addition, etc., can be applied to the stego video $V_C$. The stego video $V_C$ is then transmitted to the receiver. A trained secret message extractor is capable of mapping each frame of the stego video $V_C$ back to the secret message $M'$. As illustrated in Fig. 2, our approach comprises four modules: Information Encoding Module, Secret Message Embedding Module, Attacking Layer, Secret Message Extraction Module.

### 3.2 Information Encoding Module

The information encoding module consists of three parts: The first is identity extractor ($E_{id}$) which is responsible for extracting the specific feature representation tailored for the reference image ($I_R$).

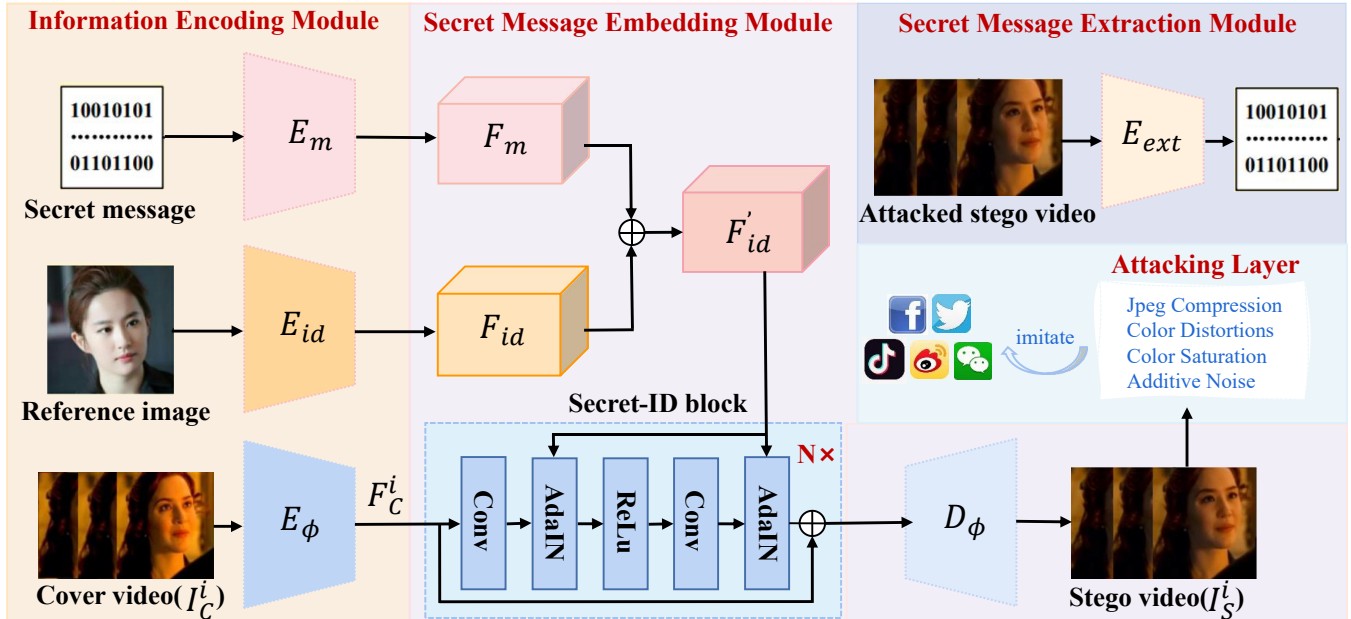

**Figure 2: The Framework of the Proposed RoGVSN.** $E_m$ **is secret message encoder.** $E_{id}$ **is identity feature extractor.** $E_\phi$ **is video feature extractor.** $D_\varphi$ **represents a video decoder.** $E_{ext}$ **represents secret message extractor. The discriminator is omitted for simplicity.**

In the proposed method, we leverage the facial recognition network [10] as the identity extractor. The second is video feature extractor $\left(E_\phi\right)$. It acquires the latent representation of cover video $V_C$ with $v$ frames, employing an encoder [7] for video feature extraction. The third is secret message encoder ($E_m$) which is a multilayer perceptron (MLP) with one dense layer. The above three parts are formulated as follows,

$$F_{id} = E_{id}(I_R) \tag{1}$$

$$F_C^i = E_\phi(I_C^i) \tag{2}$$

$$F_m = W_m M + b_m, \tag{3}$$

where $I_C^i$ denotes the $i$-th frame image of the cover video. $F_C^i \in \mathbb{R}^{C \times H \times W}$ represents the latent feature representation of $i$-th frame image. $F_{id} \in \mathbb{R}^{1 \times 512}$ denotes the identity feature matrix of the reference image. $M \in \mathbb{R}^{1 \times m}$ is the secret messages. $W_m \in \mathbb{R}^{m \times 512}$ and $b_m \in \mathbb{R}^{1 \times 512}$ denotes the learnable weights and biases.

### 3.3 Secret Message Embedding and Extraction Module

The purpose of the secret message embedding module is to embed the secret message during the face swapping scenario. The key problem is how to implement face swapping under the guidance of secret messages. To the best of our knowledge, the latent features of cover video contain the identity feature and attribute feature. The essence of face swapping is that the identity of cover video is replaced with that of reference image. As a result, we embed the secret messages into the identity feature of reference image, which

is formulated as follows,

$$F'_{id} = F_{id} + \lambda \cdot F_m, \tag{4}$$

where $\lambda$ is a hyper-parameter adjusting the influence of secret message on identity feature.

Since the identity and attribute features are highly coupled so we cannot directly extract attribute feature from the latent feature representation $F_c^i$ extracted by $E_\phi$. To better preserve the attributes, we design a Secret-ID block that consists of the modified version of the residual block [16] and the adaptive instance normalization [20] to inject $F'_{id}$ into $F_c^i$. The Secret-ID block is formulated as follows,

$$AdaIN(F_C^i, F'_{id}) = \sigma_{F'_{id}} \frac{F_C^i - \mu(F_C^i)}{\sigma(F_C^i)} + \mu_{F'_{id}}, \tag{5}$$

where $\mu(F_C^i)$ and $\sigma(F_C^i)$ represent the channel-wise mean and standard deviation of the input feature $F_C^i$, respectively. Meanwhile, $\sigma_{F'_{id}}$ and $\mu_{F'_{id}}$ correspond to two variables derived from the secret-identity feature $F'_{id}$.

After N Secret-ID blocks, the identity feature in $F_c^i$ is replaced by $F'_{id}$ and then we get $F_S^i$. Subsequently, we use an video decoder $D_\phi$ to recover the $i$-th frame $I_S^i$ of the stego video from $F_S^i$. The decoder $D_\phi$ contains four upsample blocks, a ReflectionPad layer and a convolutional layer. Each upsample block consists of a upsample layer, a convolutional layer and a BatchNorm layer. The process to get $I_S^i$ can be expressed as $I_S^i = D_\phi(F_S^i)$.

We design an extraction module to extract secret messages from the generated stego videos. The module contains seven convolutional layers with ReLU activation function. Finally, a sigmoid activation function and a binarization are added then the embedded

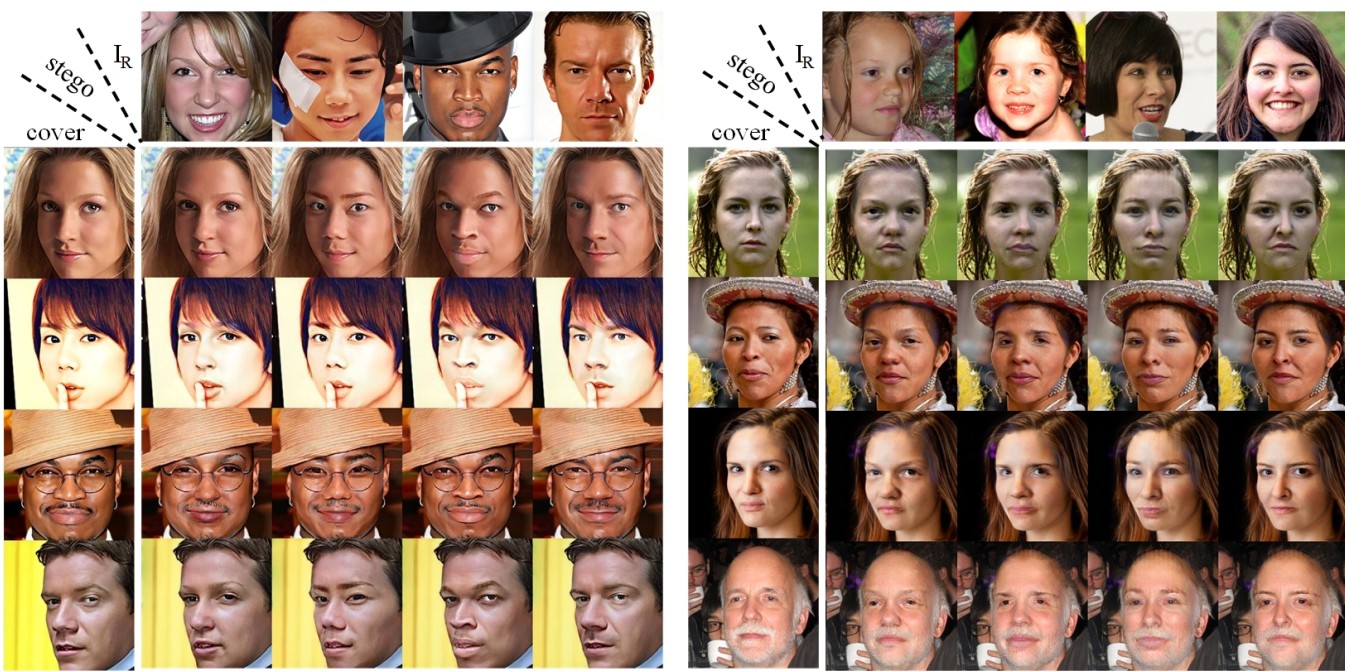

**Figure 3: Exampled Generated Stego Video Frames. Left: Vggface2. Right: FFHQ.**

secret messages is extracted. The secret message extraction module can be formulated as follows,

$$M' = E_{ext}(V_S),  \qquad (6)$$

where $M'$ is the extracted secret messages. $E_{ext}(\cdot)$ denotes the secret message extraction module. $V_S$ is the stego video.

### 3.4 Attacking Layer

To bolster the robustness of our method for the stego videos in real-world scenarios, we design an attacking layer. This module simulates prevalent distortions encountered across social network platforms, including but not limited to compression artifacts, noise interference. By subjecting the stego videos to these common distortions, our approach are more resilient to the challenges posed by real-world transmission environments. This proactive approach ensures the embedded secret message remains intact and retrievable even in the presence of various forms of degradation.

**JPEG compression.** JPEG compression is a common image compression method used in the process of photo storage. However, the quantization step is a non-differentiable because of the rounding process. To address this issue, we adopt the approach introduced by Shin et al. [32] for approximating the quantization step near zero with the function Eq. (7):

$$q(x) = \begin{cases} x^3, & |x| < 0.5 \\ x, & |x| \geq 0.5 \end{cases},  \qquad (7)$$

where $x$ denotes a specific pixel of the input image. We uniformly sample the JPEG quality from within the range of [50, 100].

**Color distortions.** We consider two general color distortions: brightness and contrast distortions that are generally brought by transmission on the Internet and the artificial processing on videos.

For the two distortions, we can perform a linear transformation on the pixels of each channel as the formula Eq. (8).

$$p(x) = a \times f(x) + c,  \qquad (8)$$

where $p(x)$ refers to the distorted and $f(x)$ refers to the original image. $a$ and $c$ are hyper-parameters to control contrast and brightness, respectively.

**Blur.** Blur is a common distortion yielded by the video processing software and the transmission on social network platforms. To simulate blur, we randomly select an angle and generate a linear blur kernel with a width ranging from 3 to 7 pixels.

**Saturation.** We perform random linear interpolation between RGB image and its grayscale equivalent to simulate the distortion.

**Noise.** We use Gaussian noise to simulate any other distortions that are not considered in the distortion module. We employ a Gaussian noise model (sampling the standard deviation $\delta \sim U[0, 0.2]$) to simulate imaging noise.

### 3.5 Loss Function

The proposed method not only ensures the quality of generated stego videos, but also guarantees the accurate extraction of secret messages. Therefore, we leverage four loss functions and a penalty to train the proposed modules, containing identity loss, attribute loss, adversarial loss, secret loss, and gradient penalty.

**Identity Loss.** The Identity Loss aims to constrain the difference between the identity feature ($F_{id}$) of reference image and the identity feature ($\hat{F}_{id}^i$) of the $i$-th frame image in generated stego video. This constraint minimizes the identity modification introduced by the secret messages, consequently enhancing the quality of the generated stego video. In this part, we leverage cosine similarity to

calculate the difference, which is expressed by the formula Eq. (9).

$$\mathcal{L}_{id} = 1 - \frac{F_{id} \times \hat{F}_{id}^i}{||F_{id}||_2 ||\hat{F}_{id}^i||_2}, \tag{9}$$

**Attribute Loss.** We use the weak feature matching loss [7] to constrain attribute difference before and after embedding secret messages. This loss utilizes the Discriminator [34] to extract multiple layers of features from the ground truth image and the generated result. The loss function is defined as Eq. (10).

$$\mathcal{L}_{att} = \sum_{j=h}^{H} \frac{1}{N_j} ||D_j(I_S^i) - D_j(I_C^i)||_1, \tag{10}$$

where $D_j$ represents the feature extractor of Discriminator D for the j-th layer, $N_j$ signifies the number of elements within the j-th layer, and $H$ denotes the total number of layers. $I_S^i$ stands for the $i$-th frame of the generated stgeo video $V_S$, while $I_C^i$ refers to the corresponding ground truth image. Additionally, $h$ denotes the layer from which we initiate the computation of the weak feature matching loss.

**Adversarial Loss and Gradient Penalty.** In the proposed method, to enhance the quality of the generated stego videos, we utilize a discriminator [34] during the training process. We adopt the Hinge version [3, 25, 28] of the adversarial loss defined as Eq. (11):

$$\mathcal{L}_{adv} = E_x[-logD(x)] + E_z[log(1 - D(z)], \tag{11}$$

where $D$ denotes the Discriminator, $x$ and $z$ in our method is respectively $I_R$ and $I_S^i$

Furthermore, we incorporate the Gradient Penalty term [2, 15] to effectively counteract the risk of gradient explosion within the Discriminator. The Gradient Penalty is in Eq. (12).

$$\mathcal{L}_{GP} = E_z[(|| \nabla_z D(z)||_2 - 1)^2], \tag{12}$$

where $z$ is the $i$-th frame $I_S^i$ of the stego video.

**Secret Loss.** The process of secret message extraction can be framed as a binary classification problem. To address this, we employ the Binary Cross-Entropy loss (BCE), which is articulated in Eq. (13). In this equation, $x$ represents the ground truth secret, while $\tilde{x}$ denotes the extracted secret.

$$\mathcal{L}_{bce}(x, \tilde{x}) = x \log(\tilde{x}) + (1 - x) \log(1 - \tilde{x}), \tag{13}$$

Lastly, we employ the secret loss to effectively regulate the process of secret message extraction. This loss function is defined by Eq. (14).

$$\mathcal{L}_{sec} = \mathcal{L}_{bce}(M, M') \tag{14}$$

**Total loss.** The total loss of our method is defined as follows,

$$\mathcal{L} = \alpha_1 \mathcal{L}_{id} + \alpha_2 \mathcal{L}_{att} + \alpha_3 \mathcal{L}_{sec} + \mathcal{L}_{adv} + \alpha_4 \mathcal{L}_{GP} \tag{15}$$

where $\alpha_1 = 10$, $\alpha_2 = 10$, $\alpha_3 = 15$, and $\alpha_4 = 10^{-5}$.

# 4 EXPERIMENTS AND DISCUSSIONS

## 4.1 Experimental Setups

**Datasets.** We use Vggface2 [4] for training and FFHQ [21] for validation. We crop and resize facial areas to a fixed $224 \times 224$ resolution for input images. To analyze quality and performance,

we randomly select 100 videos from DeepFake MNIST+ [19] to evaluate the performance.

**Implementation Details.** We train the model to encode a binary message of length $m = 9$ or 18 bits in a frame. During training, we employ Adam optimizer with a learning rate of $4 \times 10^{-4}$ and a batch size of 4. The parameters $C, W, H$ in $F_C^i$ are $C = 512$, $W = 28$, $H = 28$. The networks train for 1 million steps. After 800k steps, we introduce all attack types proposed by the Attacking Layer. The purpose of this approach is to ensure both the visual quality and robustness of the generated stego videos. We use an NVIDIA GeForce RTX 3090 GPU for our experiments. To mitigate such phenomena, we incorporate the concept of adversarial training [8, 21, 25], employing the Discriminator to discern outcomes with noticeable errors. We adopt the patchGAN [39] version of the Discriminator.

**Evaluation Metrics.** We employ Bits Per Frame (BPF), quantifying the bits number of secret message per frame in the stego video. To assess robustness, we evaluate secret message extraction accuracy under various scenarios. For security assessment, we use three steganalysis methods [23, 31, 42] to demonstrate our method's anti-detection capability.

**Baselines.** To ensure fair comparison, we align HiDDeN and LSB to this capacity. For the HiDDeN method, we use its own noise layer, in which we employ two types of noise: JPEG compression and resize (0.7-0.8). During the training phase, we embed 18 bits secret messages into per image to train the final model. This method allows each pixel to conceal up to 3 bits of data. For the LSB method, we exclusively modifies the least significant bits of the initial six pixels across the three channels. Consequently, our capacity equates to 18 bits for each frame within each test video. In addition, we embed secret message according to PWRN's paper [22]. Due to its PU based design, the capacity of PWRN is limited to 15BPF when adjusting the size of the input image to $224 \times 224$.

## 4.2 Performance Analysis

We compare the performance of our RoGVSN with image-level steganography including HiDDeN [45] and LSB [6] and video-level steganography including PWRN [22].

**Video Quality Assessment.** Fig. 3 shows qualitative results on the integrity of generated stego video frames. We perform tests within and across datasets, each containing 16 test samples. The left set of 16 samples are test images from the within VGGFace2 dataset, while the right set of 16 samples are cross-dataset test images from the FFHQ dataset. The generated faces effectively change individual identities while retaining attributes like expressions and poses. Fig. 4 illustrates the visual effects of certain intermittent frames within the stego videos. Fig. 4 shows that our method achieves visual editing by embedding secret message while injecting the identity information of the reference images into the cover videos. Furthermore, the results of the stego video frames on the right three columns exhibit excellent visual quality.

We also employ ID Similarity and VMAF two metrics to quantitatively assess the quality of stego videos produced by our model. Qualitative and quantitative analysis results in Table 3. The ID Similarity gauges the resemblance of identity feature between the generated stego video and the reference facial image. This entails calculating the cosine value between the two identity features. A

**Table 1: Comparison Results on Extraction Accuracy. "-" means "Without Distortion". (·) represents Bits Per Frame (BPF). Under different distortion scenarios, our method demonstrates superior performance in comparison.**

| Method | - | PNG | Resize (0.5) | Bit Error | Brightness | Contrast | H.264 ABR | H.264 CRF | Motion Blur | Rain | Saturate | Shot Noise |
|---|---|---|---|---|---|---|---|---|---|---|---|---|
| HiDDeN [45] | 0.9633 | 0.8342 | 0.6516 | 0.7543 | 0.7939 | 0.7813 | 0.7901 | 0.7813 | 0.7635 | 0.7624 | 0.7927 | 0.6310 |
| LSB [6] | **1.0000** | 0.4988 | 0.4932 | 0.4533 | 0.4685 | 0.4985 | 0.4921 | 0.4932 | 0.4935 | 0.5085 | 0.4885 | 0.5012 |
| PWRN [22] | **1.0000** | 0.8473 | 0.6392 | 0.8082 | 0.7959 | 0.4470 | 0.7430 | 0.7907 | 0.6004 | 0.7255 | 0.7743 | 0.8291 |
| **Ours (9)** | 0.9737 | 0.9650 | 0.8510 | 0.9393 | 0.9409 | 0.8959 | 0.8792 | 0.9566 | 0.9414 | 0.9374 | 0.9521 | 0.9059 |
| **Ours (18)** | 0.9942 | **0.9665** | **0.9486** | **0.9565** | **0.9605** | **0.9544** | **0.9634** | **0.9642** | **0.9587** | **0.9623** | **0.9612** | **0.9588** |

**Table 2: Comparison Results on Extraction Accuracy at Severity Level 2. The symbol "-" means "Without Distortion". Under different distortion scenarios, ours method demonstrates superior performance.**

| Method | - | PNG | Resize (0.5) | Bit Error | Brightness | Contrast | H.264 ABR | H.264 CRF | Motion Blur | Rain | Saturate | Shot Noise |
|---|---|---|---|---|---|---|---|---|---|---|---|---|
| Hidden [45] | 0.9633 | 0.8342 | 0.6516 | 0.7768 | 0.7945 | 0.7224 | 0.7662 | 0.7885 | 0.7662 | 0.7652 | 0.7661 | 0.6298 |
| LSB [6] | **1.0000** | 0.4988 | 0.4932 | 0.4976 | 0.4892 | 0.5011 | 0.5001 | 0.4897 | 0.4971 | 0.5014 | 0.4978 | 0.5010 |
| PWRN [22] | **1.0000** | 0.8473 | 0.6392 | 0.8113 | 0.8246 | 0.3619 | 0.7528 | 0.7395 | 0.5877 | 0.7980 | 0.7733 | 0.8678 |
| Ours (9) | 0.9737 | **0.9650** | 0.8510 | 0.9100 | **0.9283** | 0.8507 | 0.7884 | 0.8823 | **0.8992** | 0.9380 | **0.9582** | 0.8607 |
| Ours (18) | 0.9908 | 0.9648 | **0.9175** | 0.9038 | 0.8426 | **0.8845** | **0.8595** | **0.9207** | 0.8783 | **0.9410** | 0.9450 | **0.8805** |

**Table 3: Quantitative Stego Video Quality Analysis. Evaluation Metrics: ID Similarity and VMAF. The higher value indicates higher performance. (·) represents Bits Per Frame.**

| Our method | ID Similarity | VMAF |
|---|---|---|
| RoGVSN(9) | 0.8030 | 90.13 |
| RoGVSN(18) | 0.8015 | 90.11 |

higher cosine value indicates greater identity similarity and improved face-swapping effects. The VMAF primarily utilizes three indicators: visual quality fidelity (VIF), detail loss measure (DLM), and temporal information (TI). VIF and DLM pertain to spatial features within a single frame, while TI deals with temporal correlations across multiple frames. VMAF values range from 0 to 100, with higher values denoting superior quality of the stego video. The reference videos are pure face-swapping videos without secret.

**Comparisons on Extraction Accuracy & Robustness.** We conduct comprehensive experiments involving various types of lossy operations. The distortions employed in our evaluation, depicted in Fig. 5, encompass shot noise, motion blur, contrast adjustments, brightness modifications, saturation alterations, and the application of weather filters. Besides, in practical scenarios, video compression techniques are widely utilized in mobile applications to achieve seamless real-time playback. Hence, we also test the performance under two video compression operations: ABR (Average Bit Rate) compression and CRF (Constant Rate Factor) compression for the H.264 format. Each distortion except Resize has two severity levels, with each level corresponding to distinct parameters within the actual function. Higher severity level indicates more pronounced distortion effect. Table 6 presents the parameter details of the implementation. The parameter in the second column provides explanations for distortion types and operations. The quantitative comparison results in terms of accuracy are reported in Table 1

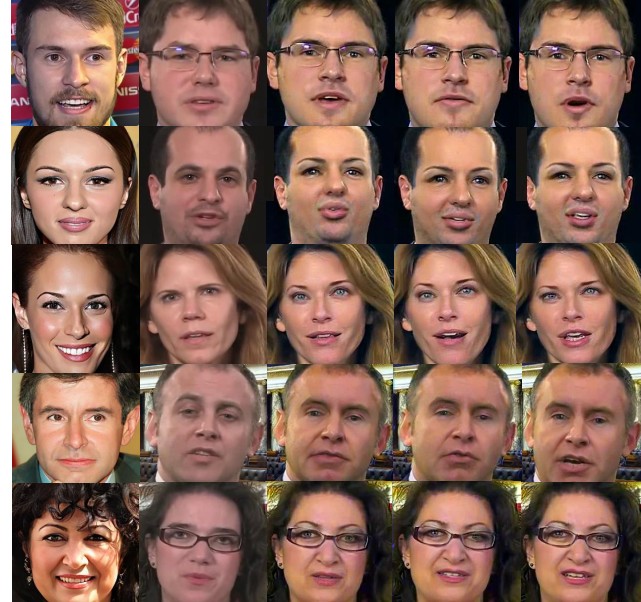

| Reference image | Frame1 of cover video | Frame1 | Frame5 | Frame10 |
| | | | Visual-editing video | |

**Figure 4: Qualitative Analysis of Stego Videos. Reference image provides identity feature as semantic feature. Column 2 represents frames within the cover videos. Frames on the right side are from the generated visual-editing stego videos.**

on severity level 1. Table 2 presents the experimental results of extraction accuracy whe the severity level is 2. The results show that our method can successfully extract secret message with high accuracy even after severe distortions. LSB [6] struggles even with PNG (quantization) and HiDDeN [45], though trained with a distortion module, can not generalize well to video-level distortions.

**Table 4: Ablation Study on Different Embedding Positions of Secret Message. Evaluation Metric: Accuracy**

| Method | - | PNG | Resize (0.5) | Bit Error | Brightness | Contrast | H.264 ABR | H.264 CRF | Motion Blur | Rain | Saturate | Shot Noise |
|---|---|---|---|---|---|---|---|---|---|---|---|---|
| Ours (a) | 0.9908 | 0.9648 | 0.9175 | 0.9425 | 0.9393 | 0.9437 | 0.9436 | 0.9514 | 0.9407 | 0.9405 | 0.9556 | 0.9241 |
| Ours (b) | 0.9479 | 0.8750 | 0.7224 | 0.8405 | 0.8452 | 0.7902 | 0.7671 | 0.8584 | 0.8480 | 0.8380 | 0.8495 | 0.8202 |
| Ours (c) | 0.9430 | 0.9388 | 0.8563 | 0.8918 | 0.8968 | 0.8624 | 0.8448 | 0.8942 | 0.8615 | 0.8763 | 0.8848 | 0.8932 |
| Ours (d) | **0.9942** | **0.9665** | **0.9486** | **0.9565** | **0.9605** | **0.9544** | **0.9634** | **0.9642** | **0.9587** | **0.9623** | **0.9612** | **0.9588** |

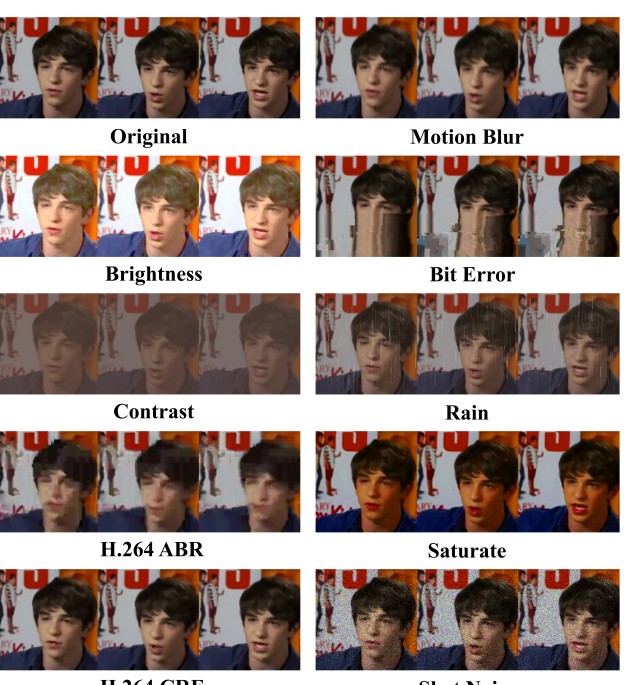

**Figure 5: Illustrative Instances of Different Distortion Operations. For each example, we extract three frames from the corrupted video using uniform sampling, with a sampling interval of three frames.**

**Table 5: Quantitative Security Analysis. Evaluation metric: AUC. Closer to 0.5 indicates higher performance.**

| Detection method | HiDDeN | LSB | PWRN | ours |
|---|---|---|---|---|
| Zhai et al. [42] | 0.5312 | 0.5423 | 0.5456 | **0.5245** |
| Li et al. [23] | 0.5416 | 0.5467 | 0.5411 | **0.5178** |
| Sheng et al. [31] | 0.5309 | 0.5189 | 0.5167 | **0.5146** |

PWRN [22] demonstrates robustness across numerous distortions, yet its performance remains constrained under operations such as motion blur or contrast adjustment. Our RoGVSN shows superior robustness to these distortions while maintaining high extraction accuracy.

**Table 6: Implementation Details of Distortions. The values under 1 and 2 respectively represents the parameters corresponding to each distortion of Level 1 and Level 2.**

| Distortions | Parameter | Severity | |
|---|---|---|---|
| | | 1 | 2 |
| Shot Noise | Photons number | 60 | 25 |
| Rain | (Density, Length) | (65,10) | (65,30) |
| Contrast | Difference portion | 0.5 | 0.4 |
| Brightness | Addition in HSV space | 0.1 | 0.2 |
| Saturate | Manipulation in HSV space | (0.3,0) | (0.1,0) |
| Motion Blur | Number of correlated frames | 3 | 5 |
| ABR | Rate of bit rate | 2 | 4 |
| CRF | Sane value | 27 | 33 |
| Bit Error | Bit error ratio | 1 / 100000 | 1 / 50000 |

**Table 7: Deepfake Detection. Evaluation Metric: AUC.**

| Detection method | No-embedding | RoGVSN |
|---|---|---|
| CADDM [12] | 0.8514 | 0.8403 |
| AltFreezing [36] | 0.9958 | 0.9923 |

**Security Analysis.** We use three video steganalysis tools to evaluate the security of our method. Li et al. [23] utilize the recompressed quantity change ratio of 25 kinds of Prediction Units (PUs) in the P slices as the 25-dimensional feature. Sheng et al. [31] extract 6-dimensional features from the rate of change in the number and ratio of $4 \times 4$, $8 \times 8$, and $16 \times 16$ PUs following I-slice recompression. Zhai et al. [42] propose to leverage low motion vector (MV) consistency within overlay videos, which exhibits high detection accuracy across multiple steganographic videos based on PU mode and MV. After constructing the above features for video steganalysis, we choose the support vector machines (SVMs) as the classifier. For each steganalysis test, the detection accuracies are averaged over 20 iterations. The detection performance of these three steganalysis schemes is presented in Table 5. It demonstrates that our method exhibits the best security compared to the three counterparts.

We also employ two state-of-the-art deepfake detection methods to assess the impact. The experimental evaluation is conducted on a dataset consisting of 300 videos, comprising 100 real videos and 100 fake videos with and without steganography embedding. Specifically, "No-embedding" and "RoGVSN" separately denote the cover videos and the stego videos. The results from the evaluation are summarized in Table 7. The findings suggest that our proposed

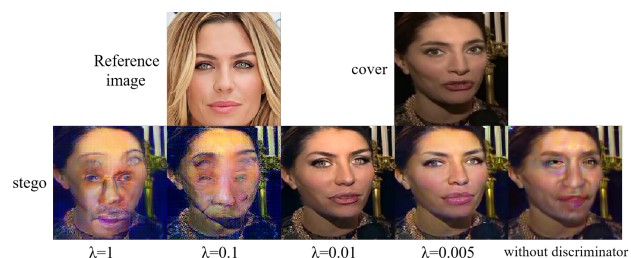

**Figure 6: Visual quality. Ablation study for $\lambda$ and discriminator. $\lambda = 0.01$ when without discriminator.**

method has minimal impact on the performance of deepfake detection. This implies that even with the embedding of secret messages using our method, the ability of deepfake detection algorithms to distinguish between real and fake videos remains largely unaffected.

### 4.3 Ablation Study

**Embedding Position of Secret Message.** Within our generation network, consisting of 9 Secret-ID blocks, we aim to examine varying positions for embedding the secret message. We partition the secret message into two 9-bit segments and determine their respective positions, as follows,

- Setting (a): 1st-4th blocks and 5th-9th blocks.
- Setting (b): 1st-2nd blocks and 3rd-4th blocks.
- Setting (c): 5th-6th blocks and 7th-8th blocks.
- Setting (RoGVSN): 1st-3rd blocks and 4th-6th blocks.

The detection performance across these four configurations is presented in Table 4. Setting (b) and (c) exhibit a substantial reduction when compared to those of Setting (a) and Setting (RoGVSN), implying that augmenting the quantity of injected secret information blocks enhances performance. Moreover, Setting (c) surpasses the performance of Setting (b), potentially due to the greater impact of subsequent blocks on the final generated image.

**Ablation on Attacking Layer.** Fig. 7 compares the results under two training scenarios: with and without distortion module. We can observe that even without distortion module, our method still possesses a certain degree of robustness, achieving extraction accuracies surpassing 0.86, which already exceeds the performance of the three comparative methods. Upon the application of distortion, the accuracy is on average improved by 6%.

**Ablation Results on $\lambda$ and the Discriminator.** The balance between visual quality and extraction accuracy is controlled by the hyperparameter $\lambda$. Selecting an appropriate value for $\lambda$ is crucial. We conduct ablation experiments on $\lambda$. Fig. 6 shows the visual quality of stego videos and Table 8 shows the extraction accuracy of secret message when $\lambda$ is different. These results indicate that setting $\lambda$ to 0.01 can achieve a satisfactory balance between visual quality and extraction accuracy. We also conduct ablation experiments on discriminator. Fig. 6 indicates the introduction of discriminator significantly enhances visual quality of stego videos.

**Table 8: The quantitative ablation study for $\lambda$ on secret message extraction accuracy.**

|  | $\lambda = 1.00$ | $\lambda = 0.1$ | $\lambda = 0.01$ | $\lambda = 0.005$ |
|---|---|---|---|---|
| Accuracy | 0.8414 | 0.5885 | **0.8607** | 0.5160 |

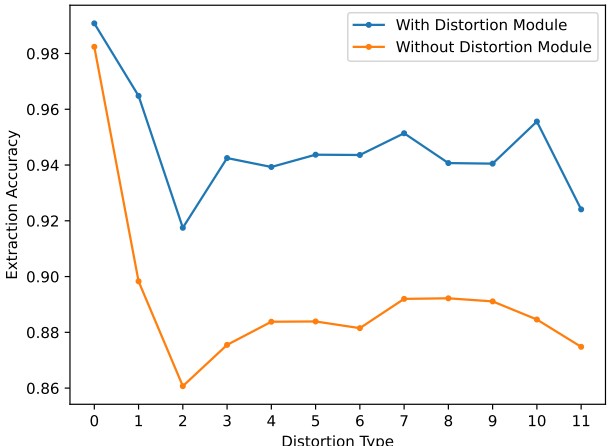

**Figure 7: Ablation Results on Attacking Layer. The horizontal axis represents distortion types, corresponding to the order listed in Table 1.**

### 5 CONCLUSION AND DISCUSSION

In this paper, we propose a method for covert transmission in online social network. Experimental results demonstrates our approach can generate high-quality stego videos and ensure accurate extraction of secret message even when subjected to various forms of distortions. The flexibility of model training makes it difficult for attackers to detect our secret message. Traditional video steganography methods leverage redundancy in video data for hiding. Although these methods have certain security and embedding capabilities, they notably lack robustness. This deficiency makes the hidden secret message susceptible to loss during compression encoding and channel transmission. Our method based on visual editing of videos can be seen as a process of modifying the semantic feature of objects within the videos. Instead of hiding secret messages in covert space, we embed them into the semantic feature of the video during visual editing. Advanced semantic feature is less susceptible to distortion, rendering this method inherently robust. Additionally, we introduce an attacking layer, which further enhances the robustness of our method.

The RoGVSN is jointly trained with four modules, containing information encoding module, secret message embedding model, attacking layer, and secret message extraction module. Experimental results of RoGVSN method applied to facial video datasets demonstrate its superiority over existing video and image steganography techniques in terms of both robustness and generalization capacity. Our work is a preliminary exploration of the robust generative video steganography. In the future, we will comprehensively investigate the robustness of generative video steganography.

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
