# OpenReview forum: "From Covert Hiding To Visual Editing: Robust Generative Video Steganography"
_acmmm.org/ACMMM/2024/Conference — MM2024 Poster_

### Official Review · Reviewer_F5Te · 2024-05-20

**Rating:** 4
**Confidence:** 2

**Summary:**

This paper introduces a novel video steganography technique that entails replacing facial features through video editing, also known as "Deepfake for steganography". Since this approach primarily emphasizes semantic features over low-level video or image features, it demonstrates a robust capability to withstand typical distortions and malicious attacks.

**Strengths:**

* A novel method of using facial editing techniques to transmit covert messages.
* Strong resilience to various distortion and malicious attacks, especially in typical social media scenarios.
* Sufficient experiments to demonstrate the efficacy and security of the proposed RoGVSN.

**Limitations:**

a) The security of the proposed method is severely compromised by the imperceptibility of Deepfake-like methods, posing an inevitable threat to the application of this method. The results in Table 7 also have indicated that the proposed RoGVSN faces a significant challenge that could be easily perceived as "abnormal". The detection of AI-generated content videos will become more common and easier in the future. Maintaining imperceptibility for facial editing is still challenging. This is the inherent drawback and may require further improvements.

b) This method has limited application as it can only process videos of humans. It would be more valuable if this method could be applied to a broader range of video types, including natural landscapes, news reports, sports events, and short videos, which are more prevalent.

c) Some other questions:
1. Details of the extraction algorithm are somewhat unclear. Is the extraction process an end-to-end classifier that directly extracts the secret message?
2. Distortion of the "Bit Error" type appears to severely impact the readability of the entire frame. How does the RoGVSN manage to survive such a chaotic situation when a significant portion of the frame image is corrupted?
3. The impact of $\lambda$ in Table 8 does not appear to follow a straightforward one-peak pattern. Results for $\lambda=1.0,0.01$ demonstrate notably superior performance compared to the other scenarios. Is there any explanation for this?
4. Steganalysis for stega-videos is somehow missing. Though this approach may differ from the previous method, it is still intriguing to determine if the created stego-videos could deceive the traditional steganalyzer to some extent.

**Suitability:**

3

---

### Official Review · Reviewer_356Y · 2024-05-24

**Rating:** 6
**Confidence:** 4

**Summary:**

This paper proposed an end-to-end robust generative video steganography network, RoGVSN. The proposed method achieves visual editing by modifying semantic features of videos to embed secret message for enhanced robustness.

**Strengths:**

1. The writing is clear, and the review of related studies is sufficient.
2. Adequate experiments and analysis demonstrate that proposed method is capable of generating high-quality stego videos and outperforms existing methods in terms of robustness and capacity.

**Limitations:**

1. Please illustrate the motivation for embedding the secret message in identity features rather than in other attribute features.
2. There are some typo errors, e.g., “ where 𝜆 is a hyper-parameter…” should be “where 𝜆 is a hyper-parameter…”.
3. Add the following reference:
Meng L, Jiang X, Sun T, et al. A Robust Coverless Video Steganography Based on the Similarity of Inter-Frames[J]. IEEE Transactions on Multimedia, 2023.

**Suitability:**

3

---

### Official Review · Reviewer_hWHL · 2024-05-24

**Rating:** 3
**Confidence:** 3

**Summary:**

The paper proposes an innovative approach to video steganography that embeds secret messages within the semantic features of videos, rather than modifying the covert space for embedding as in traditional methods.
It introduces an end-to-end robust generative video steganography network (RoGVSN) that achieves visual editing by modifying the semantic features of videos to embed secret messages.
The paper uses face-swapping as an illustrative example to demonstrate the visual editing effects.

**Strengths:**

1. It shows innovation embedding secrets within the semantic features of videos, rather than the conventional approach of modifying the covert space.
2. It shows robustness against common distortions encountered on online social networks.

**Limitations:**

1. Although this method uses face-swapping technology for data hiding against video compression, how can this steganography make a positive societal impact? From my understanding, this method modifies the semantic features and thus can not be used for the copyright protection of the original video. Face-swapping is also a technology that can be easily used for malicious editing. Does this method show any benefit against the abusive use of face-swapping technologies?
2. The novelty is limited. The face-swapping pipeline is basically the same as sim-swap, and the secret message is embedded by a simple summation in the feature space. This is basically an easy fusion of the secret message embedding and the identity embedding.

**Suitability:**

2

---

### Official Review · Reviewer_MV1s · 2024-05-25

**Rating:** 6
**Confidence:** 3

**Summary:**

The paper proposes a  generative video steganography method (RoGVSN) which generates a stego video, by embedding a secret message using semantic features extracted from a reference image.  As experimentation, The robustness of RoGVSN is evaluated using ID Similarity and VMAF.

Extraction accuracy is quantified using BPF (Bits Per Frame), compared with image level steganography methods, HiDDeN and LSB, and video level steganography method PWRM. This evaluation of extraction accuracy encompasses employing distortions, under two severity levels, and the application of video compression techniques.

For security analysis, the paper uses three video steganalysis tools to compare the AUC of RoGVSN with the same steganography methods used in the extraction accuracy evaluation. Furthermore, the AUC of cover video and stego videos of RoGVSN, under the deepfake detections mehods, CADDM and AltFreezing, are evaluated.

Finally, An ablation study is conducted to examine the  performance of the method across varying positions for embedding the secret message, across 4 different configurations.

**Strengths:**

An innovative study of steganography from the niche of utilising deepfake.

Fig.2 of the framework clearly explains the different aspects of the methodology.

**Limitations:**

The payload of hidden secret messages is limited (9 or 18bits). There is very little discussion.

The explanation of the actual meaning needs to be more convincing, beyond the social platforms. Meaningful application scenarios can be described in more depth.

The dataset on the selection of secret message and the reference images lacks details.

**Suitability:**

3

---

### Meta-Review · Area_Chair_sApN · 2024-07-02

**Recommendation:** Accept (Poster)
**Confidence:** 5

**Metareview:**

The submission received mixed reviews (accept, weak reject, accept, borderline accept). The main contribution in this work is the presentation of a generative video steganography method. A majority of reviewers regarded this work as a novel addition to existing steganography literature, with clear technique contribution and empirical evaluation. However, a few reviewers also pointed out several key weaknesses such as limited applications. Based on all reviews and correspondence, the ACs would recommend to accept this work, meanwhile suggesting the authors to revise it accordingly.